# Differences in drug intake levels (high versus low takers) do not necessarily imply distinct drug user types: Insights from a new cluster-based model

Diego M. Castaneda, Martin O. Job⦿*

Department of Biomedical Sciences, Cooper Medical School of Rowan University, Camden, New Jersey, United States of America

* job@rowan.edu

## Abstract

### Background

A current model categorizes drug takers into high versus low takers (HT and LT) based on their drug intake levels, with the assumption that these groups represent different phenotypes. When several drug doses are considered, the inverted u-shaped dose-response curves (IUDR) of HT are shifted upwards and rightward, relative to that of LT. However, these IUDR 'shifts' are not quantitative metrics and may be subjective. Also, differences in intake levels do not necessarily imply distinctions in other variables (such as demand elasticity) that are important for drug user phenotypology. With supporting evidence from a recent report, we hypothesized that, contrary to assumptions in the field, HT and LT do not necessarily represent distinct phenotypes.

### Methods

Male Sprague Dawley rats (n = 12) self-administered different doses of cocaine, and we obtained IUDR and demand curves per individual. We developed a new model to quantify the variables that defined the structure of the IUDR and we employed behavioral economic principles to obtain variables that defined the demand curve. We conducted principal component analysis/gaussian mixtures model clustering of variables from both IUDR and demand curves, to identify/compare the clusters that were revealed to HT/LT groups that were distinguished via median split.

### Results

The cluster-based model identified groups more distinct than LT versus HT. LT and HT were composed of mixtures of individuals from these distinct clusters. LT/HT were not very different when several other variables were considered.

**Data availability statement:** All relevant data are within the manuscript and its Supporting information files.

**Funding:** This work was funded by the Department of Health and Human Services/National Institutes of Health/National Institute on Drug Abuse/Intramural Research Program, Baltimore, Maryland, United States of America (to MOJ). This work was also supported by the Francis Lax Fund for Faculty Development at Rowan University, Glassboro, New Jersey, United States of America (to MOJ). This work was supported by startup funds from the Cooper Medical School of Rowan University, Camden, New Jersey, United States of America (to MOJ). The funders played no role in the study design, data collection and analysis, decision to publish and preparation of the manuscript.

**Competing interests:** The authors have declared that no competing interests exist.

## Conclusions

Differences in drug intake levels (HT versus LT) do not necessarily imply distinct phenotypes.

---

## Introduction

There is an epidemic of psychostimulant use disorders with far-reaching socioeconomic impact yet there are no FDA-approved pharmacotherapeutic intervention strategies for managing this health problem. Not all psychostimulant users are the same. To understand psychostimulant use disorders, we must understand what differentiates types/phenotypes of psychostimulants. An understanding of different drug user types may lead to more individualized or group-centric options to address the epidemic of substance use disorders.

One of the current preclinical models distinguishes high versus low takers (HT and LT) based on differences in their drug intake levels, with the assumption that these groups represent different phenotypes. There is some face-validity to this idea. For example, subjects that consume higher levels of drugs also tend to manifest higher drug seeking and display higher likelihood of relapse after withdrawal [1–3]. Also, while HT and LT may be distinguished based on differential intake *under normal conditions*, punishment-sensitive (shock-sensitive, SS, assumed to be non-compulsive) and punishment-resistant (shock-resistant, SR, assumed to be compulsive) drug takers are distinguished based on drug intake levels *under stressful conditions* such as footshock punishment [4–32].

However, there are several problems associated with the current method of drug phenotypology using intake as the distinguishing metric. First, the grouping strategy - the median split analysis of drug intake levels is typically employed to classify drug user types into HT and LT. This analytical procedure categorizes responses above and below the median response as high and low responses, respectively. However, median split analysis can be a problematic approach particularly with continuous data [33–47]. Furthermore, classifying subjects into high versus low responders does not allow for more than two groups, limiting our ability to identify groups of subjects that do not easily fit into these categories. Furthermore, complex behaviors include several co-occurring variables, and it is unclear if the median split of a single variable (out of several other variables) is an effective strategy for identifying distinct groups.

Second, the effects of psychostimulants (and other drugs) on intake levels are not linear but are defined by an inverted u-shaped dose-response curve (IUDR) (S1 Fig). Because of this structure, subjects that are designated as HT at one dose may be designated as LT at another dose (S1 Fig). To effectively identify HT versus LT, it is important to assess the entirety of the IUDR. This has already been assessed, and the assumption in the field is that the curves of HT are shifted upwards and rightward, relative to that of LT (Fig 1A, 1B).

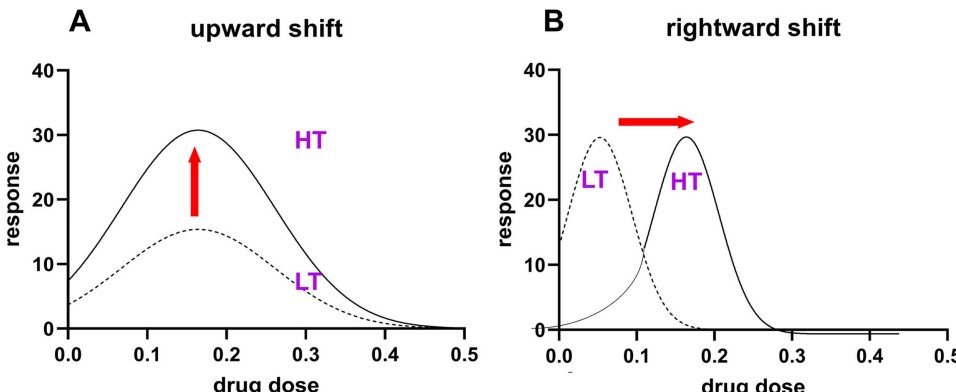

**Fig 1. The inverted u-shaped dose-response curve (IUDR) of High Takers (HT) are proposed to be shifted upward and rightward relative to the IUDR of Low Takers (LT).** A represents an upward shift while B represents a rightward shift. Note that these descriptors (upward, rightward) are not quantitative. Note also that for Graph A, the responses of HT>LT at every dose. Interestingly, for Graph B, the responses of HT>LT at higher doses whereas the responses of LT>HT at lower doses.

However, by using qualitative terms like 'rightward' and 'upward', it is clear that these IUDR 'shifts' have not been described directly using quantitative metrics. These descriptors may be subjective, and it is unclear if the IUDR of HT are quantitatively distinct from that of LT.

Third, the distinction between high versus low takers is typically achieved with respect to an intake level variable: HT take higher levels of drug than LT, though not always, see Fig 1B. However, there are other variables that are not necessarily related to intake levels, such as demand elasticity, that may distinguish different phenotypes. Demand elasticity, which is thought to be important for drug user typology [48–57], is a measure of how motivated a subject is to defend its intake under cost/price constraints. The intake of a drug is not necessarily related to the demand elasticity [52,54,58,59] in the same way that appetitive and motivational components of reinforcer intake are not the same [60,61]. This means that, in theory, despite expressing distinctions in drug intake levels, HT and LT may have similar/different demand elasticity. Thus, while LT and HT may be distinguishable based on intake levels, they may not be distinguishable based on demand elasticity. It is unclear if HT and LT are distinct groups when other variables are also considered.

Because of the aforementioned problems (median split, single variable, IUDR structure, other variables unrelated to intake variables), it is unclear that HT and LT necessarily represent different phenotypes. To move the field forward, it is important to address these problems and to offer better alternatives for understanding different drug user types/ phenotypes.

To address the problem of median split of a single variable, we can employ unbiased clustering of several variables. Because of the problem of multicollinearity associated with the assessments of several (related) variables per subject, we can employ clustering of principal components which do not have the problem of multicollinearity. Principal components are derivatives that contain information from all variables combined.

To address the problem associated with the IUDR structure, we developed a new quantitative model we term the **Qu**antitative **S**tructure of **C**urve **An**alytical (**QSCAn**) model for the IUDR (QSCAn-IUDR) (Fig 2A). The QSCAn-IUDR employs a non-linear regression using Gaussian fit to obtain variables that quantitatively define the structure of the IUDR. With this new model, we can quantify any 'shifts' in the IUDR.

To address the problem associated with the exclusion of other variables unrelated to intake variables (demand elasticity), we included analysis of the demand curve. Conceptualizing different doses as different prices, the IUDR can also be employed to derive a demand curve (Fig 2B). Thus, from the IUDR, we can simultaneously obtain IUDR structural

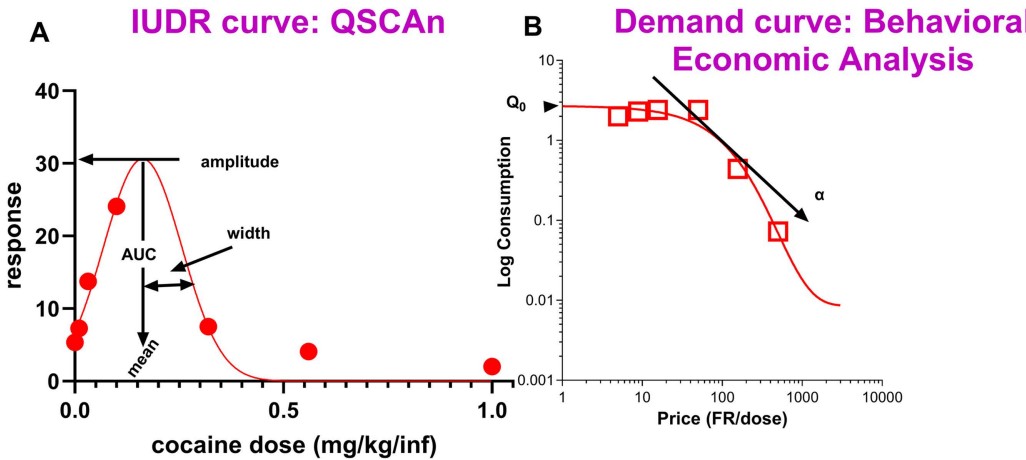

**Fig 2. The new Quantitative Structure of Curve Analytical (QSCAn) model for assessments of the IUDR and behavioral economic analysis for the assessment of the demand curve.** Graph A is a plot of dose on the x-axis and response on the y-axis to derive the IUDR. Graph A includes a gaussian fit (non-linear regression) which reveals 4 variables that define the structure of the IUDR: amplitude, mean, width and AUC. Graph B is a demand curve derived from graph A. Graph B is a plot of price (Fixed Ratio/dose) on the x-axis and log consumption on the y-axis. Graph B includes an exponential fit to reveal 2 variables ($Q_0$ and α corresponding to consumption at zero price and demand elasticity, respectively). The variable eValue (essential value) is derived from α.

variables and demand curve variables including elasticity. Thus, clustering of the principal components derived from both IUDR and demand curve variables should address all the problems (mentioned above) associated with the current methods employed to identify different types of drug users.

In a recent study, we identified distinct behavioral types of opioid users using gaussian mixtures model clustering of principal components (PCA-GMM) derived from several variables [62]. In that study, it was revealed that subjects assigned to differential drug access groups (short-access and long-access) similar to LT and HT groups were not necessarily distinct phenotypes. In line with those findings, we hypothesize that the groups, if any, that will be revealed by this careful analysis will likely be composed of mixtures of individuals from HT and LT groups – this is a challenge to the assumption that HT and LT individuals necessarily represent distinct groups.

To test our hypothesis, we allowed male Sprague Dawley rats to self-administer several doses of cocaine. We obtained 1) the response(s) per dose per individual, 2) the IUDR per individual, and 3) the demand curve per individual. We employed QSCAn-IUDR and behavioral economic analysis to obtain several variables from the IUDR and demand curves, respectively. We conducted PCA-GMM clustering of the principal components derived from the combination of the IUDR and demand curve variables to identify distinct groups of drug users, if any. We compared the groups identified via the cluster-based model with groups identified via current grouping strategies. To do this, we distinguished HT and LT based on median split of amplitude and mean (Fig 1A, 1B) and then determined if these groups were (also) distinct with respect to several other variables, and different from the groups identified via the cluster-based model. Our methods, results, discussion and conclusion are below.

## Methods and materials

### Subjects

Twelve male Sprague Dawley rats were obtained from Charles River Laboratories (Wilmington, MA). They were housed under temperature- and humidity-controlled conditions, and they were maintained on a 12-h/12-h light/dark cycle (lights on at 7:00 AM and off at 7:00 PM). After acclimation to the animal housing facility for at least one week, the weights of

rats were maintained at approximately 350 g via food restriction. Care and use of the animals was in accordance with the guidelines of the National Institutes of Health and the National Institute on Drug Abuse Intramural Research Program Animal Care and Use Program (NIDA IACUC). The facility in which the research was carried out was accredited by AALAC International.

### Animal health

Throughout the experiments from the arrival of the rats at the housing facility through surgery to the end of behavioral experiments, we ensured that rats were not in any pain or distress. After the completion of all behavioral experiments, the rats were euthanized via isoflurane anesthesia followed by decapitation according to established guidelines.

### Surgery

Chronic indwelling catheters (RJV-1, SAI Infusion Technology, Lake Villa, IL) were inserted under anesthesia (Isoflurane) using aseptic techniques into the external jugular vein of the rats. One end of the catheter was placed in the jugular vein, and the other end was attached to a backmount/external port (313-000BM-10-5UP, P1 Technologies, Roanoke, VA) that exited from the back of the rat enabling access for flushing or intravenous infusions. This exit was closed with a screw cap (C313CAC, P1 Technologies, Roanoke, VA). Meloxicam was administered as an analgesic after surgery and during post-surgery recovery. To minimize the likelihood of infection and the formation of clots or fibroids, the catheter was flushed daily with 0.2 mL of a sterile solution containing heparin (30.0 IU/mL) and the antibiotic enrofloxacin (5 mg/kg). Rats were allowed to recover from surgery for a minimum of 1 week before cocaine self-administration protocols were initiated.

### Self-administration

The operant-conditioning chambers for self-administration are as described in a previous paper [56]. All subjects were initially trained to self-administer a natural reinforcer (a 20-mg sucrose pellet) under a fixed ratio 1-response (FR1, one active lever press = response delivery) schedule in 1-h daily sessions. The pellet deliveries were followed by a time out (TO) of twenty (20) seconds during which there was no sucrose pellet delivery and during which all lights in the operant chamber were off. After the subjects had acquired and maintained this behavior, the response requirement was changed to a FR5 schedule.

Thereafter the subjects were allowed to self-administer cocaine (0.32 mg/kg/infusion) in 2-h daily sessions until they acquired and maintained this behavior. Afterwards, still on the FR5 schedule, the daily sessions were divided into five 20-min components, each separated by a 2-min TO. Then the subjects self-administered five different doses of cocaine per daily session (within-session design) with a single cocaine dose self-administered per component. To cover all the doses of cocaine, the subjects had ten daily sessions on the within session design where they administered in five 20-min components, in order, cocaine dose = d(dose) 0.032, d0.10, d0.32, d0.56, d1.00 mg/kg/infusion followed by eleven daily sessions where they administered, in order, cocaine dose = d0.00, d0.01, d0.032, d0.1, d0.32 mg/kg/infusion. The same concentration of cocaine (1.78 mg/mL) was placed in the syringes and the doses above were delivered by changes in delivered volume. For cocaine dose = 0, there was no solution delivered.

### Variable estimations and calculations

Current model: As mentioned above, each subject self-administered different doses of cocaine (cocaine dose = 0.032, 0.10, 0.32, 0.56, 1.00 mg/kg/infusion) in ten daily sessions and different doses of cocaine (cocaine dose = 0.0, 0.01, 0.032, 0.1, 0.32 mg/kg/infusion) in eleven daily sessions. We obtained the average number of infusions (responses) per cocaine dose for every subject.

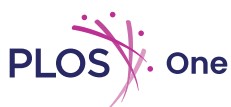

QSCAn-IUDR: From the average responses per dose for each subject, we constructed the IUDR curves which were a plot of $\log_{10}$[dose] on the x-axis and $\log_{10}$[response] on the y-axis. We employed the gaussian fit (equation 1) through which we estimated behavioral variables that defined the IUDR structure (amplitude, mean, width and AUC, see variables/variable definition in Fig 2A and S1 Fig).

$$Y = A * e^{\left(-0.5 * \left(\frac{x-x0}{b}\right)^2\right)} \tag{1}$$

where Y = log[response], X = log[dose], A = log[amplitude], x0 = log[mean], b = width. To find the values of A and mean, we used the antilog function 10^(log[A]) and 10^(log[mean]), respectively.

To obtain Area under the curve (AUC), we employed the following equation:

$$AUC = A * \sqrt{2\pi} * \ln(width) \tag{2}$$

where A = amplitude, π = pi (3.142)

Behavioral economic analysis of the IUDR-derived economic demand curve: From the IUDR curves, we obtained the demand curves. Demand curves were plotted with consumption (mg/kg) = number of reinforcers earned (infusions) × dose (mg/kg/infusion) on the y-axis and price = FR/dose (mg/kg/infusion) on the x-axis (Fig 2B). To obtain $Q_0$ (consumption at zero price) and α (demand elasticity – which is related to the decline in consumption with unit increases in price), we employed behavioral economic analysis of the IUDR-derived demand curve using the inverse of cocaine dose as a price (Fig 2B) [52,60,63,64].

The demand curve was fitted using the exponential function [65] (Fig 2B) as shown below:

$$\text{Log } Q = \text{Log } Q_0 + k \left(e^{-\alpha(Q_0 C)} - 1\right) \tag{3}$$

where Q represents consumption of the reinforcer, C represents cost, $Q_0$ represents consumption at no cost, α represents demand elasticity and is a fitted parameter related to the decline in consumption with increased cost, and k is a scaling constant reflecting the consumption range.

The demand elasticity-α is inversely related to how much work the subject is willing to do to defend consumption when prices are increased. We calculated the essential value (eValue) which is directly related to demand elasticity using the formula below:

$$eValue = \frac{1}{(100 \times \alpha \times k^{1.5})} \tag{4}$$

where α and k are as described above in equation 3.

Thus, we obtained a total of six (6) variables (Fig 2A, 2B): four from the IUDR curve structure (amplitude, mean, with, AUC) and two from the IUDR-derived economic demand curve ($Q_0$, eValue).

Principal Component Analysis: Principal component analysis (PCA) was employed to reduce variables to principal components to understand the contribution of each variable to total variability within the data set. The PCA included a standardization of all variables before deriving the principal components.

PCA-GMM clustering analysis: We conducted gaussian mixtures model (normal mixtures) clustering analysis of principal components derived from the six IUDR and demand curve variables to determine if there were any distinct groups of subjects. The criteria for group distinction was as follows: 1) the centers of the clusters identified must be clearly distinct with respect to principal component 1 (PC 1) – the linear combination of variables accounting for most of the variability

within the data set, and 2) the clusters identified must not overlap in 2-dimensional (2-D) space described by PC 1 and PC 2 and in 3-D space described by PC 1, PC 2 and PC 3. To confirm that we had the optimal number of clusters, we plotted the optimal number of clusters on the x-axis and the Bayesian Information Criterion (BIC) on the y-axis: the optimal number of clusters would correspond to the lowest BIC on this plot.

Distinguishing HT versus LT: Median split analysis: We conducted median split of all subjects at all average responses at each dose of cocaine self-administered. We also conducted median split analysis at the level of each variable derived from the IUDR curve and IUDR curve-derived economic demand curve.

Model comparison: We compared the groups identified via PCA-GMM of principal components derived from all 6 variables of the IUDR and demand curves (see variables in Fig 2A, 2B), with HT and LT groups identified via median split of IUDR amplitude and mean (see the reasons for selecting these specific variables in Fig 1A, 1B).

Power analysis: There are no standard approaches for conducting *a priori* power analysis for data-driven clustering analysis [66].

## Statistical analysis

We analyzed our data using the following software products: GraphPad Prism v 10.6.1 (GraphPad Software, San Diego, CA), SigmaPlot 14.5 (Systat Software Inc., San Jose, CA) and JMP Pro v 18 (SAS Institute Inc., Cary, NC). Data are expressed as mean ± SEM. We employed Grubb's test to identify outliers. Statistical significance was set at $P < 0.05$. For IUDR curve structure analysis for individuals, we plotted log[response] versus log[dose] per individual and employed gaussian fit of the IUDR curves to estimate amplitude, mean, width and AUC (see equations 1 and 2). For behavioral economic analysis/graphing of demand curves, we employed GraphPad templates from https://ibrinc.org/behavioral-economics-tools/. We employed multivariate analysis to determine the relationships between all the variables obtained. We employed PCA to reduce several variables into principal components and we determined the weight of the variables (IUDR and demand curve) with regards to variability in the data (how the variables loaded into the principal components). We employed PCA-GMM clustering to identify distinct clusters, if any. We utilized median split of the variables to determine HT versus LT and determined if these groups were different for several other variables and also how these groups compared to the groups identified using the cluster-based model. For analysis of IUDR and demand curves for identified/categorized groups, we employed regression analysis and 2-way repeated measures ANOVA to determine if there were any main effects of group and dose, and if there were any interactions: group × cocaine dose (0, 0.01, 0.032, 0.10, 0.32, 0.56 and 1.0 mg/kg/infusion).

## Results and discussion

The Methods we employed are described above. All our raw data are included in the Supporting Information S1 File. For every subject, the average responses per dose are shown in Table 1.

For every subject (n = 12), the IUDR and demand curves are shown in S2 and S3 Figs, respectively. Grubb's test detected one outlier (ratID K's $Q_0$ was 8.5, while the mean $Q_0$ was 3.46667 ± 0.51188) which was excluded from further analysis. The IUDR variables and demand curve variables for the remaining n = 11 subjects are shown in Table 2. Note that variables were normally distributed (S4 Fig).

Multivariate analysis: We had 13 variables per individual: 7 average response variables from 7 doses, 4 variables for the IUDR and 2 variables for the demand curves. We employed multivariate analysis to determine relatedness, or the lack thereof, of the variables to each other (Fig 3).

IUDR variable-Amplitude was significantly correlated with AUC ($P < 0.0001$), d0.01 ($P = 0.0253$), d0.032 ($P = 0.0006$, and eValue ($P = 0.0008$). IUDR variable-mean was significantly correlated with AUC ($P = 0.0131$), d0.00 ($P = 0.0253$), d0.01 ($P = 0.0037$) and d0.032 ($P = 0.0024$). IUDR variable-width was significantly correlated with AUC ($P = 0.0066$), d0.00 ($P = 0.0052$) and d0.01 ($P = 0.0294$). IUDR variable-AUC was significantly correlated with amplitude (see above), mean (see above), width (see above), d0.00 ($P = 0.0092$), d0.01 ($P = 0.0054$), d0.032 ($P = 0.0124$) and eValue ($P < 0.0001$).

**Table 1. Raw values of average responses per cocaine dose per subject.** We observed that for different doses, there were inconsistencies in the group composition of high versus low responders (based on median split of specified average responses per dose). The bolded values correspond to the high responders.

| Cocaine dose | 0.00 | 0.01 | 0.032 | 0.1 | 0.32 | 0.56 | 1 |
|---|---|---|---|---|---|---|---|
| ratID: A | 5.363636 | 7.272727 | 13.7619 | **24.09524** | 7.52381 | 4.1 | 2 |
| B | 5.545455 | 7.818182 | **24.66667** | 27.2381 | **8.238095** | **4.8** | **3.1** |
| C | 3.727273 | 6.181818 | **15.80952** | 27.66667 | **11** | **5.6** | **3** |
| D | **7.727273** | **9** | **30.04762** | 16 | 5.142857 | 3.2 | 1.5 |
| E | **9.636364** | **9.818182** | 15.33333 | **28.28571** | **11.2381** | **6.1** | **4.3** |
| F | 2.444444 | 2 | 5.5 | 14.61111 | 6 | 4.111111 | **2.777778** |
| G | 5 | 6.272727 | **17** | 16.04762 | 7 | 3.8 | 1.5 |
| H | 5 | **8.363636** | 15.71429 | 19 | 6.714286 | 3.5 | 1.7 |
| I | **8.545455** | **19.45455** | **24.7619** | 19.61905 | 6.142857 | 3.3 | 1.8 |
| J | **7.181818** | **8.363636** | **20.57143** | 22.7619 | **9.238095** | **4.4** | 2.4 |
| K | **8.454545** | 7.363636 | 13.57143 | **33.42857** | **22.19048** | **11.9** | 5.9 |
| L | **6.363636** | **7.909091** | 12.42857 | **29.42857** | **9.047619** | **5.9** | **3.4** |
| Median | 5.954545 | 7.863636 | 15.7619 | 23.42857 | 7.880952 | 4.255556 | 2.588889 |

**Table 2. Raw values of IUDR and demand curve variables per subject.** We observed that for different variables, there were inconsistencies in the group composition of high versus low responders (based on median split of specified variables). The bolded values correspond to the high responders.

| Variables | Amplitude | Mean | Width | AUC | $Q_0$ | eValue |
|---|---|---|---|---|---|---|
| ratID: A | 22.54239 | **0.061376** | 0.7562 | 42.73391 | 2.700 | 3.373096 |
| B | **29.64831** | 0.059979 | **0.7878** | **58.55338** | **3.500** | **4.287834** |
| C | **26.30268** | **0.075683** | 0.776 | **51.16791** | **4.000** | **3.892034** |
| D | **27.86121** | 0.043053 | 0.7192 | 50.2326 | 2.000 | 3.614032 |
| E | **23.87811** | **0.065917** | **0.9318** | **55.77745** | **4.300** | **4.599677** |
| F | 12.61828 | **0.112099** | 0.6497 | 20.55175 | **3.100** | 1.946017 |
| G | 20.55891 | 0.056364 | 0.7349 | 37.87601 | 2.200 | 3.011693 |
| H | 21.28139 | 0.051168 | 0.7708 | 41.12233 | 2.200 | 3.328713 |
| I | **29.51209** | 0.031117 | **0.8713** | **64.46199** | 2.100 | **4.960436** |
| J | **26.18183** | 0.057544 | **0.7961** | **52.25208** | **3.100** | **4.015591** |
| L | 22.59436 | **0.070795** | **0.8479** | 48.02646 | **3.900** | **3.952847** |
| Median | 23.87811 | 0.059979 | 0.776 | 50.2326 | 3.100 | 3.892034 |

Demand curve variable-$Q_0$ was significantly correlated with the highest 4 doses d0.1, d0.32, d0.56, d1.00 (P < 0.05), but not with the lowest 3 doses. $Q_0$ was neither correlated with the IUDR variables (P > 0.05) nor with eValue (P > 0.05). Demand curve variable-eValue was significantly correlated with all IUDR variables and with the average responses due to lower, but not higher, doses of cocaine.

The average responses for the highest 4 doses were correlated with each other but not with the average responses of the lowest 3 doses. With respect to relationships between IUDR and demand curve variables, $Q_0$ was unrelated to the 4 IUDR variables while eValue was related to the 4 IUDR variables.

PCA of all variables: We conducted PCA on all 13 variables per individual. This dimensional reduction yielded several PCs (Fig 4A) including PC 1 (accounting for ~49% data variability), PC 2 (accounting for ~ 39% data variability) and PC 3 (accounting for ~ 6% data variability).



**Fig 3. Multivariate analysis of all variables obtained.** The correlation coefficients and the P values for significant relationships are shown in the graph. The green rectangle shows relationships between average responses per dose and IUDR variables. The purple rectangle shows relationships between demand curve parameters and IUDR variables. The black rectangle shows relationships between average responses per dose and demand curve parameters. Each box shows the relationship between two intersecting variables.

The variable loading(s), or the weights of each variable as it relates to PC1 (x-axis) and PC 2 (y-axis) is/are shown in Fig 4B. Defining weak-to-moderate and moderate-to-strong loading(s) as correlation strengths - 0.5 to -1.0 and 0.5 to 1.0, respectively (Fig 4B), we determined that the lower doses (d0.0, 0.01, 0.032, 0.1), amplitude, mean, width, AUC and eValue had moderate-to-strong correlations with PC 1 whereas the higher doses (d0.3, 0.56, 1.00) and $Q_0$ contributed weakly-to-moderately to PC 1.



**Fig 4. Principal Component Analysis of all variables obtained.** We conducted PCA to reduce all variables to principal components. Graph A shows the principal components on the x-axis (PCs – PC 1, PC 2, PC 3 etc.) and the proportion of variance they represent on the y-axis. Graph B is a plot of PC 1 on the x-axis and PC 2 on the y-axis with the loading strength (correlation) of each variable shown. We defined weak-to-moderate and moderate-to-strong loading(s) as correlations strengths < 0.50 and > 0.50, respectively (see strength scale below the x-axis). Graph C is a 2-D plot of variable loading on PC1 versus PC 3. Graph D is a plot of variable loading on PC 2 versus PC 3. Note that lower doses (d0.0, 0.01, 0.032, 0.1), amplitude, mean, width, AUC and eValue had moderate-to-strong correlations with PC 1 whereas the higher doses (d0.3, 0.56, 1.00) and $Q_0$ contributed weakly-to-moderately to PC 1.

PCA of IUDR and demand curve variables: We conducted PCA on 6 variables: 4 IUDR and 2 demand curve variables. This dimensional reduction yielded several PCs (Fig 5A) including PC 1 (accounting for ~67.2% data variability), PC 2 (accounting for ~ 25.9% data variability) and PC 3 (accounting for ~ 6.2% data variability). The PCs are a linear combination of all 6 variables. Fig 5B shows the variable loading, or the weights of each variable as it relates to PC1 (x-axis) and PC 2 (y-axis). Defining weak, moderate and strong loading(s) as correlation strengths < 0.25, 0.25–0.75 and > 0.75, respectively (Fig 5B), we determined that $Q_0$ contributed weakly to PC 1 (strongly to PC 2). Mean, width and amplitude contributed moderately-to-strongly to PC 1 (weakly-to-moderately to PC 2). AUC and eValue contributed strongly to the PC with the most variability within the data (PC 1) and weakly to PC 2.

PCA-GMM clustering of IUDR and demand curve variables: As mentioned earlier, all variables show normal distribution (S4 Fig) and are appropriate for normal mixtures clustering (gaussian mixtures model clustering, GMM). PCA-GMM clustering revealed 2 clusters (cluster 1, red n = 5 and cluster 2, green, n = 6, Fig 5C), clearly distinct cluster centers 1) with respect to PC 1, 2) on the 2-D and 3-D plane, and 3) non-overlapping (in accordance with distinction criteria, see





Fig 5. Principal component analysis (PCA) and PCA-gaussian mixtures model (PCA-GMM) clustering revealed two distinct clusters. For IUDR and demand curve variables, we conducted PCA. Graph A shows the principal components on the x-axis (PCs – PC 1, PC 2, PC 3 etc.) and the proportion of variance they represent on the y-axis. Graph B is a plot of PC 1 on the x-axis and PC 2 on the y-axis with the loading strength (correlation) of each variable shown. We defined weak, moderate and strong loading(s) as |correlations strengths|< 0.25, 0.25-0.75 and > 0.75, respectively (see strength scale below the x-axis). The purple circle and the arrow show the variables (AUC, eValue) that loaded most strongly into PC 1. Graph C is a 2-D plot of PC1 versus PC 2 with clusters (from PCA-GMM clustering) shown. Graph D is a 3-D plot of PC1 versus PC 2 versus PC 3 with clusters (from PCA-GMM clustering) shown. Note that clusters are clearly distinct on the 2-D and 3-D planes and are non-overlapping. Graph E are the clusters identified in C and D via PCs, with constituent variables shown. Graph F is a plot of the optimal number of clusters (= 2).

Methods section). A 3-dimensional representation shows that these clusters were clearly distinct in a 3-D space defined by PC 1 (x-axis), PC 2 (y-axis) and PC 3 (z-axis) (Fig 5D). Fig 5E shows the clusters with regards to the variables of the IUDR and the demand curves. The optimal number of clusters identified from 10 iterations of the clustering algorithm was 2 (Fig 5F).

Model comparisons: As mentioned in the Methods section, HT IUDR curves are shifted 'upward' and 'rightward' relative to LT IUDR curves. In quantitative terms, this suggests that the IUDR amplitude and mean of HT versus LT are thought to be significantly different (Fig 1A, 1B). Thus, we employed median split of amplitude to categorize subjects into LT-a (amplitude) and HT-a groups (Fig 6G–6L). Additionally, we employed median split of mean to categorize subjects into LT-m (mean) and HT-m groups (Fig 6M–6R). We compared the clusters identified via PCA-GMM to HT/LT.

Cluster1 and cluster2 were distinct (P<0.05) with respect to the averages of width, AUC, $Q_0$ and eValue (Fig 6C–6F), but not amplitude and mean (P>0.05, Fig 6A, 6B). These clusters were distinct for 4/6 variables (or ~67%). LT-a and HT-a were different for 50% of the variables assessed while LT-m and HT-m were different for 33.3% of the variables assessed. Clusters 1 and 2 are more distinct than the groups derived from median split of amplitude and mean (Fig 6).

Distinctions with respect to PC 1: Cluster1 and cluster2 were clearly distinct with respect to PC 1 and these clusters were non-overlapping (Fig 7A).

**Fig 6. The groups identified via the cluster-based model are more distinct than HT versus LT.** These graphs represent comparisons between amplitude, mean, width, AUC, $Q_0$ and eValue for cluster1 and cluster2 (Graphs A-F), LT-a (amplitude) and HT-a (Graphs G-L), and LT-m (mean) and HT-m (Graphs M-R), respectively. The P values from unpaired t-tests with Welch's correction are shown for each graph. Cluster1 versus cluster2 were the most distinct groups when all these variables were considered. HT and LT were not very distinct groups when all variables were considered.



**Fig 7. HT and LT are not very distinct and include mixtures of members from distinct clusters.** The above represents comparisons between groups identified via 1) the cluster-based model (Cluster1 and 2, Graphs A, D and G), 2) median split of amplitude (LT-a and HT-a, Graphs B, E and H) and median split of mean (LT-m and HT-m, Graphs C, F and I). Graph A shows that cluster1 and cluster2 were clearly distinct with respect to PC 1 and these groups were non-overlapping. HT-a versus LT-a were also clearly distinct with regards to PC 1 and non-overlapping (Graph B), though less distinct relative to the clusters in Graph A. Graph C shows that HT-m and LT-m were not clearly distinct with respect to PC 1 and were overlapping (but clearly distinct with respect to PC 2). Note that HT and LT (for amplitude and mean) consisted of mixtures of members from clusters 1 and 2 (compare Graphs B and C with A). Clusters were distinct with respect to IUDR and demand curve (Graphs D and G). HT and LT (for amplitude and mean) were distinct with respect to IUDR but not with respect to demand curves (Graphs E-F, H-I). Significant differences (P<0.05) between groups is shown by the symbol #. Significant differences (P<0.05) following post-hoc tests (Tukey's) is represented by the symbol *.

Differences between cluster1 and cluster2 were related to about 67% variability within the data (see # in Fig 7A), as were differences between LT-a and HT-a (Fig 7B). Even if there were any differences between HT-m and LT-m, these differences were relevant to only about 30% variability within the data set (see # in Fig 7C).

Interestingly, while also revealing differences with respect to PC 1, LT-a and HT-a were not identical to cluster1 and cluster2. Furthermore, LT-m and HT-m were not the same composition as cluster1 and cluster2 or LT-a/HT-a. LT-a (n = 5) was composed of 80% subjects (4/5) from cluster1 and 16% of the subjects (1/6) from cluster2, while HT-a (n = 6) was composed of 20% of the subjects (1/5) from cluster1 and 83% of the subjects (5/6) from cluster2 (this is from comparison between Fig 7B and Fig 7A). LT-m (n = 5) includes a combination of 60% subjects (3/5) from cluster1 and 33.3% (2/6) of the subjects from cluster2, while HT-m (n = 6) was composed of 40% of the subjects (2/5) from cluster1 and 67% (4/6) of the subjects from cluster2 (Fig 7C compared to Fig 7A).

IUDR comparison: We compared the IUDR structures of cluster1 versus cluster2 using 1) non-linear regression analysis, and 2) Two-way repeated measures ANOVA with factors being log[cocaine-dose] (6 levels) and group (2 levels). For cluster1 versus cluster2, gaussian fit (see equation 1) yielded curves that were distinct when amplitude, mean and width were all considered together (F 3, 60 = 6.439, P = 0.0007, Fig 7D). Two-way repeated measures ANOVA revealed the following information: no dose × group interaction (F 1.456, 13.10 = 0.1717, P = 0.7760), a main effect of group (F 1, 9 = 14.51, P = 0.0042), and a main effect of dose (F 1.456, 13.10 = 59.87, P < 0.0001).

As with the clusters, curves for LT-a versus HT-a were also distinct when amplitude, mean and width were all considered together (F 3, 60 = 4.923, P = 0.0040, Fig 7E). Two-way repeated measures ANOVA revealed the following information: no dose × group interaction (F 1.543, 13.89 = 1.209, P = 0.3156), a main effect of group (F 1, 9 = 5.638, P = 0.0416), and a main effect of dose (1.543, 13.89 = 65.86, P < 0.0001).

For LT-m versus HT-m, curves were also distinct when amplitude, mean and width were all considered together (F 3, 60 = 7.194, P = 0.0003, Fig 7F). Two-way repeated measures ANOVA revealed the following information: a significant dose × group interaction (F 2.049, 18.44 = 7.985, P = 0.0030), no main effect of group (F 1, 9 = 0.06675, P = 0.8019), but a main effect of dose (2.049, 18.44 = 112.9, P < 0.0001).

Demand curve comparison: We compared the structure of the demand curves of respective groups using 1) non-linear regression analysis (exponential model by [65], see equation 3). For cluster1 versus cluster2, the exponential model yielded curves (Fig 7G) that were distinct when $Q_0$ and α were both considered together (F 2, 8 = 4.8, P = 0.0422), when only α was considered (F 1, 8 = 9.2, P = 0.0164), and when only $Q_0$ was considered (F 1, 8 = 5.9, P = 0.0410). Note that the eValue is a derivative of α (see equation 4).

For LT-a versus HT-a, the exponential model yielded curves (Fig 7H) that were not distinct (for the comparison of the demand curve structure) – no differences when $Q_0$ and α were both considered together (F 2, 8 = 2.9, P = 0.1119), when only α was considered (F 1, 8 = 5.3, P = 0.0504), and when only $Q_0$ was considered (F 1, 8 = 0.77, P = 0.4052).

For LT-m versus HT-m, exponential model yielded curves (Fig 7I) that were distinct only when $Q_0$ was considered (F 1, 8 = 5.4, P = 0.0483), but these curves were neither distinct for α only (F 1, 8 = 0.053, P = 0.8244) nor for $Q_0$ and α together (F 2, 8 = 3.9, P = 0.0672).

Note that, unlike with the clusters (Fig 7D and 7G), LT versus HT (amplitude, mean) were significantly different with respect to the IUDR (Fig 7E, 7F), but these differences were not necessarily reflected in the demand curves (Fig 7H, 7I).

## Discussion

It is widely assumed that HT and LT (and other variations of these groups that are distinguished based on differential drug intake levels) represent distinct drug user types/phenotypes. However, there are several problems associated with this assumption (see Introduction section). To address these problems, in order to advance the field, we have developed analytical tools to identify distinct groups of subjects by 1) obtaining/including previously unquantified variables, and 2) conducting unsupervised clustering of principal components derived from several variables. While employing these

tools in another study [62], we determined that differential levels of drug intake in a different drug use model (similar to that observed for HT versus LT groups) did not necessarily imply distinct phenotypes. Thus, we developed a hypothesis, contrary to common assumptions, that HT and LT are not necessarily distinct phenotypes. In this study, we tested and confirmed our hypothesis.

From the average responses for different drug doses, we plotted the IUDR per subject. From IUDR, we constructed the demand curve per subject. We employed the new QSCAn-IUDR and behavioral economic analytical tools to obtain relevant variables from both plots (Fig 2). For every subject, we obtained 13 variables. Multivariate analysis showed some relatedness, and the lack thereof, between variables (Fig 3). PCA revealed that lower cocaine infusion doses (d0.0, 0.01, 0.032, 0.1), amplitude, width, AUC and eValue were the more important variables to be considered with respect to our ability to detect distinct groups, whereas the higher doses (d0.3, 0.56, 1.00), mean and $Q_0$ were less effective in this regard (Fig 4). We conducted PCA-GMM clustering of IUDR and demand curve variables – this procedure revealed distinct clusters (Fig 5). Interestingly, we determined that these distinct clusters (cluster1 and cluster2) were 1) more different from each other than were HT/LT, and 2) included mixtures of individuals from HT and LT groups (Figs 6 and 7). Conversely, HT and LT groups included mixtures of individuals from the different clusters. By showing that 1) HT and LT groups are not clearly distinct, especially when other variables (including demand curve parameters) are also assessed (Figs 6 and 7), and 2) HT and LT groups consisted of mixtures of individuals from distinct clusters (Fig 7), we confirmed our hypothesis that HT and LT do not necessarily represent distinct groups. This study corroborates a recent study from our lab [62] that revealed that subjects grouped by the experimenter as short-access (LT-like) and long-access (HT-like) are 1) not representative of distinct phenotypes and 2) are composed of mixtures of different types of drug users.

Previous studies have suggested that vertical shifts (and rightward shifts) in the IUDR curve (Fig 1A, 1B) may serve as a distinguishing feature for high versus low drug responders [1,3,67]. When we distinguished HT and LT via median split of their amplitude variable, we observed differences in the IUDRs of HT versus LT, as expected, but these differences were either not major contributors to overall variability in the data and/or these groups were not distinct with regards to demand curves (Figs 6 and 7). Thus, HT/LT groups, while different with regards to the IUDR, as proposed (Fig 1) were not clearly distinct with regards to demand curves (Figs 6 and 7).

Multivariate analysis revealed that drug intake (at zero price) was unrelated to demand elasticity (Fig 3), reinforcing the findings from other studies [52,54,58,59]. PCA revealed that higher doses (d0.3, 0.56, 1.00) and $Q_0$ were correlated, but also that these variables contributed weakly-to-moderately to variability within the data. Interestingly, in [1], HT and LT were distinguished via median split of their cocaine intake at the highest dose (1 mg/kg/infusion). Also, in [3], rats were grouped as HT and/or LT based on median split of their intake of cocaine at a dose = 0.5 mg/kg/infusion: a relatively high dose. These groups were distinguished using median split of high cocaine doses, but our data suggests that the combination of median split and high cocaine doses may not be the most effective grouping strategy.

There are some limitations to this study especially with regards to biological sex (only male), single strain (Sprague Dawley) and small power (low n). With regards to biological sex, it is plausible that our results would not apply to females [68]. However, in this outbred strain, sex differences are not very evident (relative to other rat strains). For example, there were no clear sex differences with regards to psychostimulant self-administration in the Sprague Dawley rat strain: sex differences were reported in only 4 [69–72] out of 32 studies [69–101]. We do not think that there will be sex differences with respect to our results. Unlike the Sprague Dawley rat strain, there are observed sex differences (75% of studies) in psychostimulant self-administration behavior for the Long Evans (LE) outbred rat strain [10,102–115]. It could be argued that there will be sex differences in the LE outbred strain. However, there is some evidence that differences in psychostimulant effects between males and females may not be driven primarily by biological sex [116–120]. Similar to the mixed groups of individuals from different clusters being represented in HT versus LT (Fig 7), we predict/expect that there will be a mixture of males and females in any and every cluster/group. That said, we will study sex differences in the future.

We acknowledge that there are limitations posed by the number of subjects employed in this study. More work will be done in future to address these limitations. That said, our results are supported by several studies suggesting that these limitations, while important, may not change our results overall. For example, we show that demand elasticity is an important variable to distinguish distinct groups of drug users (Figs 4–6)- this is buttressed by several other studies that have shown that α, or demand elasticity may be important variable(s) for drug user typology [48–57]. Moreover, our results with respect to HT and LT not being distinct are consistent with a previous study that had more power [62].

There are widespread implications of our study. Preclinical approaches that are based on differential drug intake as a behavioral metric to separate drug users into distinct drug use types or 'phenotypes', using current methods, may need to be interpreted carefully based on the limitations that we have mentioned above and based on our findings.

Our study may be clinically relevant: the clusters we identified were distinct with respect to both of the demand curve parameters- $Q_0$ and eValue (Figs 6E, 6F, 7G) and economic demand curve parameters are clinically-relevant and directly translatable to the human condition [50,121–123]. Our new objective approach may be important in advancing the field of drug user typology in a manner that is clinically translatable.

## Conclusions

Our work confirms major limitations of grouping drug users based on the median split of drug intake variables (Tables 1 and 2). Identifying distinct groups via unbiased clustering of principal components derived from several variables, as we have shown in this study, is a superior alternative to the median split. The IUDR of HT and LT, while different, do not imply that they belong to different drug user types/phenotypes (especially when other variables such as demand elasticity are also considered). Our findings may advance the field.

## Supporting information

**S1 Fig. Problems identifying high and low responders under an inverted U-shaped dose-response curve.** For these graphs, the x-axis represents dose of drug self-administered and the y-axis represents response. Graph A is a representation of an inverted U-shape dose-response curve (IUDR) showing some variables that can be derived from the structure. Amplitude is the peak of the IUDR, width represents how wide the IUDR is, mean is the center of the IUDR, and AUC represents the area under the IUDR. Graph B-D represent 3 examples of the IUDR of two curves (curve 1 and curve 2). In graph B, curve 2 would always have a higher response compared to curve 1 at all doses of drug self-administered. Note also that relative to curve 1, curve 2 has the same mean (center) value, but has greater amplitude, width and AUC. If curve 1 and curve 2 represent low and high responders, respectively, these group designations would be stable across all doses. However in graph C, with respect to responses, curve 1 > curve 2 at lower doses and curve 1 < curve 2 at higher doses. In graph C, curves have similar amplitude, AUC, and width but different mean values. A subject designated as high responder at one dose may be a low responder at a different dose – these group designations would not be stable across all doses: curve 2 would be designated low responders at lower doses and high responders at higher doses, and vice versa for curve 1. In graph D, curve 2 would be designated as the high responder group and curve 1 would be designated as the low responder group at higher doses, but it would be difficult to distinguish these groups at lower doses, and these groups appear to have different amplitude, mean, width and AUC. In summary, it can be challenging to identify low and high responders based on qualitative descriptions of the IUDR without quantifying the variables that define this structure. Note that the various intake patterns presented above may not cover all possible patterns.
(TIF)

**S2 Fig. The inverted U-shaped dose-response curves for all individuals.** The IUDR curves are a plot of $\log_{10}$[dose] on the x-axis and $\log_{10}$[response] on the y-axis. A-L represent the inverted U-shaped dose response curves for all individuals.

$R^2$ values (written into the graphs) were from gaussian curve fit. Rat K was excluded because it was an outlier. The amplitude, mean, width and AUC for all other individuals are shown in Table 2.
(TIF)

**S3 Fig. The economic demand curves derived from inverted U-shaped dose-response (IUDR) curves for all individuals.** A-L represent the economic demand curves for individuals. $R^2$ values (written into the graphs) were from exponential fit for demand curves. K was excluded because it was an outlier. The estimated $Q_0$ and α are shown in Table 2.
(TIF)

**S4 Fig. Variables for IUDR and demand curves are normally distributed.** A-F represent the distribution analysis of IUDR variables (amplitude, mean, width, AUC) and demand curve variables ($Q_0$ and eValue). A, D and F can be fitted using Weibull function and B, C and E can be fitted using lognormal function. These fits approximate normal distributions. We employed gaussian mixtures model in clustering analysis.
(TIF)

**S1 File. Raw data.**
(XLSX)

## Acknowledgments

The authors wish to acknowledge Dr. Jonathan L Katz in whose lab most of the experiments were carried out. Both authors contributed to data analysis and to the writing of the manuscript. MOJ designed and conducted behavioral experiments and statistical analysis.

## Author contributions

**Conceptualization:** Martin O. Job.

**Data curation:** Martin O. Job.

**Formal analysis:** Martin O. Job.

**Investigation:** Martin O. Job.

**Methodology:** Martin O. Job.

**Project administration:** Martin O. Job.

**Resources:** Martin O. Job.

**Software:** Martin O. Job.

**Supervision:** Martin O. Job.

**Validation:** Martin O. Job.

**Visualization:** Diego M. Castaneda, Martin O. Job.

**Writing – original draft:** Diego M. Castaneda, Martin O. Job.

**Writing – review & editing:** Diego M. Castaneda, Martin O. Job.

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
