## [Decision Letter · Decision Letter 0]

7 Jun 2025

Dear Dr. Job,

Thank you for submitting your manuscript to PLOS ONE. After careful consideration, we feel that it has merit but does not fully meet PLOS ONE’s publication criteria as it currently stands. Therefore, we invite you to submit a revised version of the manuscript that addresses the points raised during the review process.

We look forward to receiving your revised manuscript.

Kind regards,

Rita Fuchs

Academic Editor

PLOS ONE

Journal Requirements:

3. Thank you for stating the following financial disclosure: NIDA DA000547

5. Please expand the acronym “NIDA” (as indicated in your financial disclosure) so that it states the name of your funders in full.

The authors wish to acknowledge Dr. Jonathan L Katz in whose lab most of the experiments were carried out. Both authors contributed to data analysis and to the writing of the manuscript. MOJ designed and conducted behavioral experiments and statistical analysis. This work was funded by the Department of Health and Human Services/National Institutes of Health/National Institute on Drug Abuse/Intramural Research Program, Baltimore, MD, USA [grant -DA000547]. This work was also supported by the Francis Lax Fund for Faculty Development at Rowan University. This work was also supported by startup funds from Rowan University, Camden, New Jersey.

Please remove any funding-related text from the manuscript and let us know how you would like to update your Funding Statement. Currently, your Funding Statement reads as follows: NIDA DA000547

8. We note that you have included the phrase “data not shown” in your manuscript. Unfortunately, this does not meet our data sharing requirements. PLOS does not permit references to inaccessible data. We require that authors provide all relevant data within the paper, Supporting Information files, or in an acceptable, public repository. Please add a citation to support this phrase or upload the data that corresponds with these findings to a stable repository (such as Figshare or Dryad) and provide and URLs, DOIs, or accession numbers that may be used to access these data. Or, if the data are not a core part of the research being presented in your study, we ask that you remove the phrase that refers to these data.

9. Please remove all personal information, ensure that the data shared are in accordance with participant consent, and re-upload a fully anonymized data set.

Additional Editor Comments:

This is a thought-provoking paper that clearly moves the field forward. The relatively small sample size and limited variation (males only, single strain) introduces some weaknesses into data interpretation. These and other feedback about the limitations of the modeling approach should be acknowledged prominently.

Reviewers' comments:

Reviewer's Responses to Questions

**Comments to the Author**

1. Is the manuscript technically sound, and do the data support the conclusions?

Reviewer #1: Yes

Reviewer #2: Partly

2. Has the statistical analysis been performed appropriately and rigorously?

Reviewer #1: I Don't Know

Reviewer #2: Yes

3. Have the authors made all data underlying the findings in their manuscript fully available?

Reviewer #1: Yes

Reviewer #2: Yes

4. Is the manuscript presented in an intelligible fashion and written in standard English?

Reviewer #1: Yes

Reviewer #2: Yes

Reviewer #1: The present study employs behavior economics and cluster analyses to probe potential distinctions between individual profiles exhibited in a rat model of cocaine taking. The study has several strengths including detailed analyses of individual rat profiles, employment of cluster analyses which extend commonly employed median splits, as well as examination between different dependent variables. The manuscript is largely well written but could benefit from truncation of results in the main text in favor of a placing some data and results into a supplement. The main conclusion that median split subgrouping of intake levels, particularly determined using a single dose of drug, are unlikely to capture meaningful distinctions between subgroups/subpopulations is supported by the results—this is an important point. Overall, this study is likely to be of interest to researchers in behavioral pharmacology and neuroscience but could be improved in a number of ways.

Was any type of power analyses considered? The dataset is based on a low number of subjects (although they are examined at multiple intake doses). For comparison, the Edwards et al 2007 study employed 40 rats compared to the 11 employed here. Is there concern that there may be insufficient subject variation to capture critical features of “typology”? i.e. to identify sufficiently different subjects based on cluster analyses. Conversely, there are extensive statistical consideration of a rather limited dataset involving many comparisons between different pairs of means. Please explain why type I errors are not a concern with this approach.

Intake fitted curves (figure 4) appear to be inappropriate. Namely, fitted curves go to zero at higher doses but actual intake does not. This appears to be due to significant skew in the upper end of the dose range which is not accounted for by the modeling employed. Along these lines, visualization of data in figure 4 could benefit from use of log scale on the X-axis. In addition, the dose-response curve was incomplete (did not capture the end of the descending arm). Some discussion of this apparent discrepancy is warranted in addition, some description of the actual pattern of the curve in relation to the idealized possibilities presented in Figure 1 would seem appropriate.

Generally, the relation of alpha from demand curves only loosely corresponds to motivation and using the terms interchangeably is controversial. The authors should consider providing citation to support their contention as well as potential providing alternative interpretations.

Overall, I concur with the authors that the prior use of the term addicted (e.g. based on median split of drug intake at a single dose) is inappropriate but I feel their own usage may have challenges too. First, rats do not get addicted but they can exhibit addiction-like behavior; this is an important distinction as addiction or substance abuse disorder are clinical conditions which may be modeled in rodents. Second, the need for “all the variables of the IUDR” to exhibit differences between subgroup appears arbitrary. Why must "meaningful" subgroups differ along all dimensions? Is there clinical evidence that individuals suffering from substance abuse disorder exhibit differences along all of these dimensions or is this a strategy to describe maximally subgroup a heterogenous sample of rats? i.e. is this modeling clinical conditions or behavioral analyses? If it is the latter then the terminology of addiction or addiction-like would seem inaccurate as the authors seek to merely describe subgroups of rats. For modeling addiction, it would be important to consider the available (although limited) behavioral economic analyses of human drug taking patterns.

Moderate/minor concerns:

Conclusion in abstract should reiterate high vs low takers based on median split does not produce distinct subtypes; as noted above there are many ways to divide a group and some may produce more meaningful distinctions than others (i.e. top 10% vs bottom 10% may have utility for distinct typology categorization).

Figure 1 should note that the various intake patterns presented may not cover all possible patterns.

A lot of the data (amplitude, mean, width and AUC; Qo and Alpha) is presented in the text as well as in table 2; this is redundant. I recommend removing the data presented in the text.

There are extensive, possibly excessive, analyses of the data centered of median split for each derived variable. Much of this information should be put into a supplement and have the main findings retained in the main article results; perhaps retain the median split based on alpha (as this has its own subtitle albeit, it is unclear why this variable has its own subtitle and others do not) or just state the general findings prior to the describing the results from the clustering analyses.

Reviewer #2: This manuscript by Castaneda and Job thoughtfully challenges the common practice of using median splits of drug intake to classify psychostimulant users as "high" or "low" responders, an approach often complicated by the typical inverted U-shaped dose-response (IUDR) curve and the critical distinction between intake and motivation. The authors’ central critique of simplistic dichotomization based on median splits is timely and well-argued, representing an important contribution to discussions on methodological rigor in behavioral phenotyping. The effort to employ more sophisticated, multi-parameter methods to differentiate responder types is a commendable and important direction for the field. Using 11 male Sprague Dawley rats self-administering cocaine, they derived structural and behavioral economic variables from individual IUDR curves. The generation of such a multi-faceted dataset from individual subjects is a valuable aspect of the study design. The authors found that median splits led to inconsistent group compositions, while their clustering analyses consistently revealed only one cluster across various datasets. They conclude that "high" and "low" drug takers, as commonly defined, might not represent distinct user types and propose that true distinctions would require differences across all six of their derived IUDR and economic variables. While the conceptual framework is appealing, significant concerns with the current execution and reporting limit the study’s generalizability and the authors’ claims.

Strengths of the Manuscript

The manuscript effectively highlights the inherent instability and potential for misclassification when using median splits on single measures of drug intake, especially with complex dose-response data like IUDR curves. This is a crucial reminder for the field. The introduction of a multi-variable clustering approach based on both IUDR characteristics and behavioral economic parameters is a methodologically sophisticated step forward.

Major Comments:

1 - The study’s conclusions are drawn from a very small sample of 11 male Sprague Dawley rats sourced from a single supplier, which raises significant concerns. With such a small sample size, the statistical power to reliably detect multiple clusters, especially if the differences between them are subtle, is inherently low. Consequently, the consistent finding of "only one cluster" might reflect an inability of the analysis to resolve underlying heterogeneity due to insufficient power, rather than a definitive absence of distinct user types. Furthermore, the use of only one sex (males) and a single outbred strain (Sprague Dawley) significantly limits the potential for observing broader behavioral or biological diversity. This homogeneity is a critical issue because sex differences in response to psychostimulants are well-documented, and different genetic backgrounds (even among outbred strains from different suppliers or entirely different strains) can contribute to varied behavioral phenotypes. This restricted diversity inherently limits the generalizability of the findings and may contribute to the failure to detect distinct clusters that might be apparent in a more diverse animal population. Collectively, the small sample size and the homogeneity of the animal cohort (all male, single strain from one supplier) mean the "only one cluster" finding may very well be an artifact of these experimental limitations or specific to this particular sample, rather than a robust, generalizable conclusion about drug user typology. To strengthen the conclusions and improve generalizability, more animals should be included in these studies, ideally incorporating females and potentially animals from different vendors or strains.

2 - In the manuscript, the authors use a "normal mixtures clustering" and suggest its capability to find multiple groups. However, specific details regarding the algorithm’s parameterization, the statistical criteria used for determining the number of clusters (for example, the paper does not explicitly mention criteria like the Bayesian Information Criterion (BIC), common for mixture models, as the driver for consistently finding only one cluster), or sensitivity analyses related to these methodological choices are not provided within the methods section. Furthermore, the absence of cross-validation techniques or other robustness checks, which are also not mentioned in the methods, highlights a significant concern. Without such validation steps, it is difficult to ascertain whether the "only one cluster" finding is stable and not unduly influenced by the specific composition of this small sample or the chosen analytical parameters. Given that clustering is arguably the most important analysis in the paper, this lack of detailed information in the methods is a notable omission.

3 - The variables subjected to clustering include drug intake (cocaine self-administration responses at various doses, ranging from approximately 2 to 30 infusions as per Table 1), IUDR structural parameters (for example, "amplitude" in the range of 20–40, "mean" around 0.1–0.18, "width" around 0.08–0.13, and "AUC" around 5–9, as per Table 2), and behavioral economic parameters like Q0 (approximately 2–4.3) and alpha (a very small number, approximately 5×10^−4 to 1.3×10^−3, as per Table 2). These variables inherently possess vastly different numerical scales. The manuscript does not specify in its Methods section whether these variables were standardized before any of the clustering analyses were performed. This omission is concerning because, without appropriate scaling, variables with larger numerical ranges (like "amplitude" or raw intake counts) can disproportionately influence the underlying model fitting and distance measures in clustering algorithms. This could potentially bias the results, leading to a cluster dominated by variables with the largest variance and masking the true contribution of variables with smaller absolute values but significant theoretical importance, such as alpha.

4 - The "global clustering" approach, as described, combined observed responses at all cocaine doses with IUDR structural variables (amplitude, mean, width, AUC) and economic demand parameters (Q0, alpha). It is important to note that the IUDR parameters are derived from fitting Gaussian functions to the individual dose-response data (i.e., the observed responses at all cocaine doses), and the economic parameters (Q0, alpha) are subsequently derived from demand curves which are transformations of these same IUDR curves. This means many of the inputs into this specific global clustering algorithm are not independent. The derived parameters are mathematical transformations of the original intake data they are being clustered with. Such lack of independence could inadvertently lead to the original intake patterns (and their inherent structure or noise) being over-weighted in the clustering analysis, potentially obscuring more subtle relationships or reinforcing a particular structure based on redundant information. The rationale for this specific combined analysis requires more robust justification, particularly concerning how the statistical assumptions of the clustering method are met with non-independent variables or how potential issues of multicollinearity and variable weighting were addressed.

5 - The "normal mixtures clustering" employed in the manuscript inherently assumes that any underlying data clusters have Gaussian distribution. This is a notable assumption because if true behavioral or biological subgroups manifest with different, non-elliptical structures, or vary in density, such heterogeneity might be sub-optimally detected or missed entirely, a risk amplified by the study’s small sample size. To more comprehensively assess the data structure and bolster the robustness of the "one cluster" conclusion, exploring a range of alternative clustering algorithms would be beneficial. For instance, hierarchical clustering could provide a valuable exploratory overview, visually revealing potential nested groupings or outliers without presupposing the number of clusters. Additionally, density-based approaches like DBSCAN are designed to identify clusters of arbitrary shapes and can be more robust to noise, offering a different perspective if the data does not conform to clear, centrally condensed groups.

Minor Comments:

Line 85: There’s an extra period in "limitation."

Line 110 and Line 114: "vertical" is used (for example, "shifted vertical and rightward"). It might be more grammatically conventional to use "vertically" (for example, "shifted vertically and rightward").

Line 574: "every dose of elf-administered cocaine" should be "self-administered."

**Do you want your identity to be public for this peer review?** For information about this choice, including consent withdrawal, please see our Privacy Policy

Reviewer #1: No

Reviewer #2: No

---

## [Author Response · Author response to Decision Letter 1]

12 Dec 2025

PONE-D-24-58025

Differences in drug intake levels (high versus low takers) do not necessarily imply distinct drug user types: insights from a new cluster-based model

PLOS ONE

Dear Dr. Job,

Thank you for submitting your manuscript to PLOS ONE. After careful consideration, we feel that it has merit but does not fully meet PLOS ONE’s publication criteria as it currently stands. Therefore, we invite you to submit a revised version of the manuscript that addresses the points raised during the review process.

We look forward to receiving your revised manuscript.

Kind regards,

Rita Fuchs

Academic Editor

PLOS ONE

Journal Requirements:

Response to Editor: We have complied with PLOS ONE’s style requirements expressed in the above hyperlinks.

Response to Editor: We have included the following sentence in the Methods section: “Throughout the experiments from surgery to the end of behavioral experiments, we ensured that rats were not in any pain or distress. After the completion of all behavioral experiments, the rats were euthanized via isoflurane anesthesia followed by decapitation.

3. Thank you for stating the following financial disclosure: NIDA DA000547

Response to Editor: I have amended the funding information as follows:

Funding Information

This work was funded by the Department of Health and Human Services/National Institutes of Health/National Institute on Drug Abuse/Intramural Research Program, Baltimore, MD, USA. This work was also supported by the Francis Lax Fund for Faculty Development at Rowan University. This work was also supported by startup funds from Rowan University, Camden, New Jersey. The funders had no role in study design, data collection and analysis, decision to publish, or preparation of the manuscript."

I have also included the above in the cover letter.

Response to Editor: I have corrected this.

5. Please expand the acronym “NIDA” (as indicated in your financial disclosure) so that it states the name of your funders in full.

Response to Editor: I have corrected this. I have also included this in the cover letter.

The authors wish to acknowledge Dr. Jonathan L Katz in whose lab most of the experiments were carried out. Both authors contributed to data analysis and to the writing of the manuscript. MOJ designed and conducted behavioral experiments and statistical analysis. This work was funded by the Department of Health and Human Services/National Institutes of Health/National Institute on Drug Abuse/Intramural Research Program, Baltimore, MD, USA [grant -DA000547]. This work was also supported by the Francis Lax Fund for Faculty Development at Rowan University. This work was also supported by startup funds from Rowan University, Camden, New Jersey.

Please remove any funding-related text from the manuscript and let us know how you would like to update your Funding Statement. Currently, your Funding Statement reads as follows: NIDA DA000547

Response to Editor: We have removed all funding information from the manuscript. We will move only to funding statement and ensure that it is in accordance with Editor’s request.

Response to Editor: We had already written “Care and use of the animals was in accordance with the guidelines of the National Institutes of Health and the National Institute on Drug Abuse Intramural Research Program Animal Care and Use Program (NIDA IACUC).”

I added ‘and use’ and NIDA IACUC to the resubmission.

The ethics committee that approved the study was the NIDA IACUC

8. We note that you have included the phrase “data not shown” in your manuscript. Unfortunately, this does not meet our data sharing requirements. PLOS does not permit references to inaccessible data. We require that authors provide all relevant data within the paper, Supporting Information files, or in an acceptable, public repository. Please add a citation to support this phrase or upload the data that corresponds with these findings to a stable repository (such as Figshare or Dryad) and provide and URLs, DOIs, or accession numbers that may be used to access these data. Or, if the data are not a core part of the research being presented in your study, we ask that you remove the phrase that refers to these data.

Response to Editor: We have removed any words pertaining to ‘data not shown’. We have provided all relevant data we used in analysis – raw data in Tables 1 and 2 in the main manuscript.

9. Please remove all personal information, ensure that the data shared are in accordance with participant consent, and re-upload a fully anonymized data set.

Additional Editor Comments:

This is a thought-provoking paper that clearly moves the field forward. The relatively small sample size and limited variation (males only, single strain) introduces some weaknesses into data interpretation. These and other feedback about the limitations of the modeling approach should be acknowledged prominently.

Reviewers' comments:

Reviewer's Responses to Questions

Comments to the Author

1. Is the manuscript technically sound, and do the data support the conclusions?

Reviewer #1: Yes

Reviewer #2: Partly

2. Has the statistical analysis been performed appropriately and rigorously?

Reviewer #1: I Don't Know

Reviewer #2: Yes

3. Have the authors made all data underlying the findings in their manuscript fully available?

Reviewer #1: Yes

Reviewer #2: Yes

4. Is the manuscript presented in an intelligible fashion and written in standard English?

Reviewer #1: Yes

Reviewer #2: Yes

5. Review Comments to the Author

6. PLOS authors have the option to publish the peer review history of their article (what does this mean?). If published, this will include your full peer review and any attached files.

Do you want your identity to be public for this peer review? For information about this choice, including consent withdrawal, please see our Privacy Policy.

Reviewer #1: No

Reviewer #2: No

Response to Editor

Dear Editor

We have read through the concerns of the reviewers, and we have reviewed our manuscript to address their concerns. We want to remark that the editor’s advice and the reviewer’s concerns and how we have addressed them have led to a better manuscript.

Editor’s Advice:

This is a thought-provoking paper that clearly moves the field forward. The relatively small sample size and limited variation (males only, single strain) introduces some weaknesses into data interpretation. These and other feedback about the limitations of the modeling approach should be acknowledged prominently.

Response to Editor: We used males only (and we have now acknowledged this as a limitation) but we have included the reasons why we think that this result would be applicable to females also in our discussion.

We used Sprague-Dawley rats only (and we have now acknowledged this as a limitation) but we have included the reasons why we think that this result would be applicable to other strains.

We used a small number – n = 11. We agree that there is a relatively small sample size for males only (n = 11). We agree that this is a weakness and we have addressed this limitation in the discussion section. We have included the following in the discussion section:

Line 501-522 Discussion section

There are some limitations to this study especially with regards to biological sex (only male), single strain (Sprague Dawley) and small power (low n). With regards to biological sex, it is plausible that our results would not apply to females [68]. However, we used Sprague Dawley rats and in this outbred strain, sex differences are not very evident (relative to other rat strains). For example, there were no clear sex differences with regards to psychostimulant self-administration in the Sprague Dawley rat strain: sex differences were reported in only 4 [69–72] out of 32 studies [72–86][69–71,87–101]. Unlike the Sprague Dawley rat strain, there are observed sex differences (75% of studies) in psychostimulant self-administration behavior for the Long Evans (LE) outbred rat strain [29,102–115]. We do not think that there will be sex differences regarding our results. Moreover, there is some evidence that differences in psychostimulant effects between males and females are not driven primarily by biological sex [116–120]. Similar to the mixed groups of individuals from different clusters being represented in HT versus LT (Fig 5), our observations seem to suggest that there will be a mixture of males and females in any and every cluster/group. That said, we will study sex differences in the future.

We acknowledge that there are limitations posed by the number of subjects employed in this study. More w

---

## [Editor Report · Decision Letter 1]

18 Dec 2025

Differences in drug intake levels (high versus low takers) do not necessarily imply distinct drug user types: insights from a new cluster-based model

PONE-D-24-58025R1

Dear Dr. Job,

We’re pleased to inform you that your manuscript has been judged scientifically suitable for publication and will be formally accepted for publication once it meets all outstanding technical requirements.

Kind regards,

Rita Fuchs

Academic Editor

PLOS One
---

## [Editor Report · Acceptance letter]

PONE-D-24-58025R1

PLOS One

Dear Dr. Job,

I'm pleased to inform you that your manuscript has been deemed suitable for publication in PLOS One. Congratulations! Your manuscript is now being handed over to our production team.

Kind regards,

on behalf of

Dr. Rita Fuchs

Academic Editor

PLOS One